# Multi-Omics Profiling of Human Endothelial Cells from the Coronary Artery and Internal Thoracic Artery Reveals Molecular but Not Functional Heterogeneity

**DOI:** 10.3390/ijms241915032

**Published:** 2023-10-09

**Authors:** Alexey Frolov, Arseniy Lobov, Marsel Kabilov, Bozhana Zainullina, Alexey Tupikin, Daria Shishkova, Victoria Markova, Anna Sinitskaya, Evgeny Grigoriev, Yulia Markova, Anton Kutikhin

**Affiliations:** 1Department of Experimental Medicine, Research Institute for Complex Issues of Cardiovascular Diseases, 6 Sosnovy Boulevard, Kemerovo 650002, Russia; frolav@kemcardio.ru (A.F.); shidk@kemcardio.ru (D.S.); markve@kemcardio.ru (V.M.); cepoav@kemcardio.ru (A.S.); grigev@kemcardio.ru (E.G.); markyo@kemcardio.ru (Y.M.); 2Laboratory for Regenerative Biomedicine, Research Institute of Cytology of the Russian Academy of Sciences, 4 Tikhoretskiy Prospekt, St. Petersburg 194064, Russia; lobov@incras.ru; 3SB RAS Genomics Core Facility, Institute of Chemical Biology and Fundamental Medicine of the Siberian Branch of the Russian Academy of Sciences, 8 Prospekt Akademika Lavrentieva, Novosibirsk 630090, Russia; kabilov@niboch.nsc.ru (M.K.); tupikin@niboch.nsc.ru (A.T.); 4Centre for Molecular and Cell Technologies, Research Park, Saint Petersburg State University, 7/9 Universitetskaya Embankment, St. Petersburg 199034, Russia; st022665@student.spbu.ru

**Keywords:** endothelial cells, endothelial heterogeneity, coronary artery, internal thoracic artery, coronary artery bypass graft surgery, multi-omics, transcriptome, RNA sequencing, global gene expression, proteomic profiling

## Abstract

Major adverse cardiovascular events occurring upon coronary artery bypass graft surgery are typically accompanied by endothelial dysfunction. Total arterial revascularisation, which employs both left and right internal thoracic arteries instead of the saphenous vein to create a bypass, is associated with better mid- and long-term outcomes. We suggested that molecular profiles of human coronary artery endothelial cells (HCAECs) and human internal mammary artery endothelial cells (HITAECs) are coherent in terms of transcriptomic and proteomic signatures, which were then investigated by RNA sequencing and ultra-high performance liquid chromatography-mass spectrometry, respectively. Both HCAECs and HITAECs overexpressed molecules responsible for the synthesis of extracellular matrix (ECM) components, basement membrane assembly, cell-ECM adhesion, organisation of intercellular junctions, and secretion of extracellular vesicles. HCAECs were characterised by higher enrichment with molecular signatures of basement membrane construction, collagen biosynthesis and folding, and formation of intercellular junctions, whilst HITAECs were notable for augmented pro-inflammatory signaling, intensive synthesis of proteins and nitrogen compounds, and enhanced ribosome biogenesis. Despite HCAECs and HITAECs showing a certain degree of molecular heterogeneity, no specific markers at the protein level have been identified. Coherence of differentially expressed molecular categories in HCAECs and HITAECs suggests synergistic interactions between these ECs in a bypass surgery scenario.

## 1. Introduction

Rather than being a simple semi-permeable barrier protecting the blood vessels from thrombosis, lipid retention, and mineral deposition, endothelial cells (ECs) deploy numerous angiocrine factors including cytokines, growth factors, vasoactive substances, and other signaling molecules regulating vascular and systemic homeostasis in a juxtacrine or paracrine manner [1,2,3,4,5]. Single-cell RNA sequencing has demonstrated an organ-specific pattern of EC differentiation [6,7,8,9] and molecular heterogeneity between arterial, capillary, venous, and lymphatic ECs [10,11,12,13,14,15]. Recent advances in single-cell analysis and endothelial differentiation fostered a concept of therapeutic lymphangiogenesis, a treatment implying the guided regeneration of lymphatic networks upon the lymph node dissection to remove or ameliorate lymphoedema [16,17,18]. Isolation and culture of peripheral blood-derived, endothelial, colony-forming cells, which combine the molecular signatures of arterial, venous, and lymphatic ECs and have a high proliferation rate, significantly improved the development of tissue-engineered vascular grafts [19,20,21,22,23]. Hence, our knowledge of endothelial heterogeneity has been implemented into regenerative medicine and may be valuable for healthcare as soon as clinical trials of pre-endothelialised tissue-engineered vascular grafts or local treatment with autologous cells or angiogenic growth factors show their efficiency.

For >50 years, coronary artery bypass graft (CABG) surgery remains the most frequent cardiovascular intervention [24,25,26,27]. As compared with percutaneous coronary intervention, coronary artery bypass graft (CABG) surgery is superior for patients with severe coronary artery disease and those with diabetes mellitus [27,28,29,30,31,32]. Multiple arterial grafting (MAG), which employs the left internal thoracic artery (ITA) in combination with the right ITA and/or the radial artery as conduits for coronary artery bypass graft (CABG) surgery, is associated with better mid- and long-term outcomes in comparison with single arterial grafting (SAG) which combines the use of the left ITA and saphenous vein (SV) [33,34,35,36,37,38,39,40,41,42,43,44]. Notably, the superiority of total arterial revascularisation and bilateral ITA grafting to single ITA grafting was arguable [38,45,46,47,48,49,50], although these CABG techniques were beneficial in patients <70 years of age [40] and in women [51]. To clearly define the efficiency of all MAG modalities, a randomized comparison of the clinical outcome of single versus multiple arterial grafts (ROMA) trial has been initiated, but the results have not been published hitherto [52,53,54]. A comparison of the two approaches of ITA graft harvesting, the pedicled technique which preserves tunica adventitia and perivascular adipose tissue, and skeletonisation which dissects the ITA without any surrounding tissue, provided inconclusive results, although the former showed better long-term outcomes [55,56] and the latter was associated with lower risk of sternal wound complications [57,58,59]. The reduced rate of major adverse cardiovascular events accompanying the pedicled technique might indicate an important role of endothelial cells which assemble vasa vasorum and regulate adventitial fibroblasts and perivascular adipocytes by supplying them with context-specific biochemical cues.

The reasons behind the superiority of MAG over SAG might include a higher mechanical competence of anastomoses between the ITA, radial artery, and coronary artery because all of them are muscular arteries with a similar histoarchitecture and diameter [60,61,62,63,64,65,66,67,68,69], relatively low (≈25%) collagen content, optimal glycosaminoglycan profile and negligible protease activity in the arterial wall [70,71,72,73], quiescent profile of arterial vascular smooth muscle cells (VSMCs) [60,74,75,76,77,78,79,80,81,82,83], higher local secretion of vasodilators (e.g., nitric oxide) by ITA as compared to SV [84,85,86,87,88,89,90,91], resistance to oxidative stress [92,93,94], favourable paracrine interactions between human coronary artery endothelial cells (HCAECs) and human internal thoracic artery endothelial cells (HITAECs) [91], and better vasoreactivity of muscular arteries in response to vasoactive substances [95,96,97,98,99] in conjunction with their anti-thrombotic profile [100,101]. Among these, adequate vasodilation (largely mediated by efficient nitric oxide production without endothelial nitric oxide synthase uncoupling), release of secreted factors into the circulation, and resistance to thrombosis are furnished by ECs. Hence, upkeep of the physiological profile of HCAECs and HITAECs can be considered a key factor to achieve a favourable outcome of CABG surgery in the long term.

Earlier, we documented that HCAECs and HITAECs have mutual benefits if cultured together in vitro, whilst the interactions between arterial and venous ECs are less profitable [91]. To uncover the molecular basis of this phenomenon, we applied an unbiased multi-omics approach, employing RNA sequencing to analyse the transcriptome and ultra-high performance liquid chromatography-mass spectrometry (UHPLC-MS/MS) to perform a label-free proteomic profiling of HCAECs and HITAECs. We hypothesized that HCAECs and HITAECs demonstrate a certain extent of molecular heterogeneity that does not contradict their molecular congruence, defined as the common or synergistically upregulated biochemical pathways and the absence of distinctive functional protein markers.

## 2. Results

As proteins are central signaling molecules in the human organism and proteome analysis is of major importance for pathophysiology, we first performed label-free proteomic profiling for unbiased and quantitative analysis of proteins expressed in HCAECs and HITAECs. Bioinformatic analysis included denotation of differentially expressed proteins (DEPs, defined as those with a logarithmical fold change ≥ 1 and false discovery rate (FDR)-corrected *p* value ≤ 0.05) with further drawing of Venn diagrams and volcano plots, and pathway enrichment analysis of observed-versus-expected expression of DEPs within the molecular categories through the established bioinformatic tools (Gene Ontology, Reactome, UniProtKB, and KEGG). Having employed the mentioned workflow, we revealed 244 and 287 proteins (DEPs) upregulated in HCAECs and HITAECs, respectively, whereas 2794 proteins were not differentially expressed between these cell lines (Figure 1 and Appendix A).

Among the molecular categories, both HCAECs and HITAECs overexpressed those responsible for cell attachment to BM, organisation of intercellular junctions, energy generation in mitochondria, and release of EVs (Appendix A). However, each of the cell lines also had specific upregulated categories. For instance, HCAECs overexpressed a higher number of proteins accountable for BM formation, collagen synthesis, metabolism of lipids, organic acids, vitamins and carbohydrates, Golgi transporters, and lysosomal and peroxisomal proteins (Appendix A), while proteins related to the synthesis of mitochondrial ribosomes, mitochondrial transcription and translation, ribosomal activity, biosynthesis and metabolism of nitrogen compounds, mRNA regulation, synthesis of macromolecules, and formation of elastic fibers were overrepresented in HITAECs (Appendix A). Hence, HCAECs and HITAECs exhibited certain molecular heterogeneity.

However, only 12 proteins were overexpressed in HCAECs or HITAECs with a logarithmical fold change > 5 (PUM1, MGST1, MA2B2, A1AT, PIR, FABP4, BL1S2, and DPP4, HCAECs; CSN7B, PGTB2, RCN3, and CAMLG, HITAECs). None of these proteins have been earlier reported as a specific marker of any EC lineage, although dipeptidyl peptidase 4 (DPP4) and adipocyte fatty acid-binding protein (FABP4) are frequently referred to in cardiovascular biology literature.

To better quantify the molecular evolution of HCAECs and HITAECs, we performed transcriptomic profiling under the laminar flow and in the static conditions by means of RNA sequencing. The bioinformatic analysis was the same, with the exception of defining differentially expressed genes (DEGs) instead of DEPs. Under laminar flow, we found 1014 and 849 genes (DEGs with a logarithmical fold change ≥ 1 and FDR-corrected *p* value ≤ 0.05) upregulated in HCAECs and HITAECs, respectively, whilst 19,228 genes have not been differentially expressed between the cell lines (Figure 2 and Appendix A).

The number of DEGs was from 3- to 5-fold higher than DEPs because of post-transcriptional regulation and higher coverage of transcriptomic analysis where the sequencing depth outperforms the protein annotation algorithms after UHPLC-MS/MS. At RNA sequencing, both HCAECs and HITAECs overexpressed genes accountable for the synthesis of collagen and other ECM components, adhesion to the BM, binding to glycosaminoglycans, Ca^2+^-dependent signaling, organisation of intercellular junctions, secretion of EVs, DNA metabolism and repair, RNA processing, and protein utilisation processes such as ubiquitination and autophagy (Appendix A). However, HCAECs were enriched with transcripts responsible for biosynthesis and folding of collagen, organisation of the ECM and BM, assembly of intercellular junctions, binding of integrins to collagen and proteoglycans, and formation of elastic fibers (Appendix A), whilst HITAECs had a higher number of transcripts accountable for cytoskeleton organisation, synthesis of glycosaminoglycans, release of EVs, and pro-inflammatory signaling (Appendix A).

Under static conditions, we found 729 and 926 genes (DEGs with a logarithmical fold change ≥ 1 and FDR-corrected *p* value ≤ 0.05) upregulated in HCAECs and HITAECs, respectively, whilst 17,789 genes have not been differentially expressed between the cell lines (Figure 3 and Appendix A).

Both HCAECs and HITAECs overexpressed genes responsible for synthesis of collagen and other ECM components, adhesion of ECs to the ECM, organisation of intercellular junctions, and release of EVs (Appendix A). The expression profile of HCAECs was enriched with transcripts associated with vascular development, production of nitrogen compounds, and proteolysis (Appendix A), whilst transcripts related to endosomal and lysosomal compartments, pro-inflammatory response, synthesis of macromolecules, DNA repair, RNA processing and splicing, ribosome biogenesis, and translation were overrepresented in HITAECs (Appendix A).

Similar number of DEGs in HCAECs and HITAECs under laminar flow and at static conditions (1863 and 1655, respectively) and their distribution across HCAECs and HITAECs (HCAECs: 1046 DEGs under laminar flow and 729 DEGs at static conditions; HITAECs: 849 DEGs at laminar flow and 926 DEGs under static conditions) testified to the concordance of the results between experimental models and the higher sensitivity (but lower specificity) of RNA sequencing in comparison with proteomic profiling. Taking the results of transcriptomic and proteomic profiling together, both HCAECs and HITAECs demonstrated pronounced expression of genes and proteins responsible for the synthesis of collagen and other ECM components, cell adhesion to ECM and BM components, assembly of intercellular junctions, secretion of EVs, and metabolism of macromolecules (in particular nucleic acids and proteins). When comparing HCAECs and HITAECs, HCAECs had a higher variety and expression of molecular terms attributable for the formation of BM, biosynthesis and folding of collagen, and organisation of intercellular junctions, whereas HITAECs showed molecular signatures of active pro-inflammatory signaling, generation of nitrogen compounds, synthesis of macromolecules, regulation of RNA metabolism, and ribosome biogenesis and activity.

To verify the results obtained from transcriptomic profiling, we performed selective gene expression analysis in HCAECs and HITAECs at the baseline conditions (i.e., in serum-free medium supplied with growth factors) using reverse transcription-quantitative polymerase chain reaction (RT-qPCR). At laminar flow culture, HCAECs demonstrated reduced expression of *ICAM1*, *SELE,* and *SELP* genes which encode cell adhesion molecules for leukocyte adhesion, in line with the downregulation of genes encoding pro-inflammatory interleukins such as *IL6*, *CXCL8*, and *CCL2*, in comparison with HITAECs (Table 1). Intriguingly, *KLF2* and *KLF4* genes showed opposite expressions in HCAECs and HITAECs, which is suggestive of distinct mechanosensitive transcription factors in different EC lines. In accordance with the literature [84,85,86,87,88,89,90,91], *NOS3* gene expression was lower in HCAECs as compared to HITAECs. At static conditions, HCAECs also showed downregulation of the genes for cell adhesion molecules (*VCAM1*, *ICAM1*, and *SELE*) and chemokine genes (*CXCL8* and *CCL2*), although the expression of *IL6*, *CXCL1*, and *MIF* genes was inconsistent between the experiments and expression of *NOS3* gene was also uncertain (Table 1). In terms of relative gene expression, *ICAM1* prevailed over *VCAM1*; *CCL2* and *CXCL8* showed higher expression than *IL6*; and *KLF4* was overexpressed in comparison with *KLF2* (at laminar flow as these factors convey the mechanotransduction signals) in both HCAECs and HITAECs (Table 1). Expression of endothelial-to-mesenchymal transition transcription factors varied across the experiments and was not concordant between the cell lines (Table 1).

The findings revealed at proteomic profiling were confirmed by fluorescent Western blotting for the proteins corresponding to some of the genes which expression has been measured at RT-qPCR, followed by total protein normalisation to confirm the equal protein loading across the bands. In keeping with the gene expression measurements, VCAM1, TWIST1, and KLF4 expression was higher in HITAECs as compared with HCAECs, whilst other transcription factors of endothelial-to-mesenchymal transition (Snail, Slug, and ZEB1), ICAM1, and eNOS showed minor if any expression differences (Figure 4A–G).

The results of selective gene expression measurements by RT-qPCR and protein measurements by Western blotting were concordant with those documented by whole transcriptomic sequencing and proteomic profiling, respectively. For instance, HITAECs were notable for molecular signatures of upregulated pro-inflammatory signaling (i.e., production of cytokines and chemokines, including interleukin-6) at the transcriptomic profiling (i.e., RNA-seq) and for increased expression of *CXCL8* and *CCL2* genes (encoding interleukin-8 and monocyte chemoattractant protein, respectively) at RT-qPCR, both at laminar flow and in static culture conditions. Further, HITAECs showed proteomic signatures of enhanced generation of nitrogen compounds and *NOS3* gene was evidently higher expressed at laminar flow. As HCAECs and HITAECs also exhibited minor differences in protein expression at Western blotting, we suggest that RT-qPCR and Western blotting results provide additional evidence for the molecular congruence between these EC lines.

As the concept of reciprocal and beneficial paracrine effects of HCAECs and HITAECs implies multiple interactions between their DEPs and DEGs, we carried out a functional enrichment analysis of protein–protein and gene–gene interaction networks using STRING database and Cytoscape software with the stringApp plugin. Bioinformatic analysis pipeline included: (1) filtration of DEPs and DEGs having ≥1 interaction (254 DEPs, 516 DEGs at laminar flow, and 466 DEGs under static conditions); (2) colour mapping in order to distinguish interacting DEPs and DEGs between HCAECs and HITAECs; (3) pathway enrichment analysis using GO, Reactome, UniProtKB, and KEGG terms; and (4) selection of such molecular terms relevant for arterial homeostasis.

Having employed this workflow, we found that interacting DEPs and DEGs in HCAECs and HITAECs were overrepresented in pathways (i.e., molecular terms) responsible for the maintenance of endothelial monolayer and basement membrane (various types of intercellular junctions as well as cell–cell and cell–matrix adhesion), as well as in angiogenesis, EC proliferation, elastic fiber formation, and nitric oxide synthesis pathways (Figure 5, Figure 6 and Figure 7 and Appendix A). Most of the molecular terms were in concord between the proteomic and transcriptomic data, as well as between the transcriptomic data obtained under laminar flow and at static conditions. Hence, we found that the dataset of interacting DEPs and DEGs between HCAECs and HITAECs showed significant enrichment in arterial homeostasis pathways that suggests another argument towards beneficial interactions between these two EC lines.

To reinforce the study findings, we examined the data from 102 patients who underwent repeated coronary angiography 10 years after CABG surgery. Out of 311 bypasses, we documented 88 bypass dysfunctions (i.e., adverse outcomes), and the proportion of adverse outcomes was significantly higher when using SV grafts (54/160, 33.75%) as compared with utilising left or right ITA (34/151, 22.52%, *p* = 0.038, Appendix A). Hence, the relative contribution of SV graft dysfunction into the total adverse outcomes upon the CABG surgery was 61.36% (54 out of 88 adverse outcomes), in comparison with only 36.66% when using ITA conduits (34 out of 88 adverse outcomes, Appendix A). Among the causes of bypass dysfunctions, atherosclerosis and graft degeneration (triggered by endothelial dysfunction) were responsible for 42 out of 54 adverse outcomes (77.78%) in SV conduits but had been accountable only for 9 of 34 adverse outcomes (26.47%) in ITA grafts where competitive flow (a surgical complication) prevailed (23 of 34 adverse outcomes, 67.65%, Appendix A). Therefore, bioinformatic and molecular biology evidence for the synergy between HCAECs and HITAECs were concordant with the clinical observations showing the superiority of ITA conduits as compared with SV grafts.

## 3. Discussion

Although molecular heterogeneity between different EC lineages [10,11,12,13,14,15] and ECs in distinct organs of the human body is well established [6,7,8,9], its pathophysiological and clinical significance remain unclear. CABG surgery, which is a seminal technique to treat coronary artery disease, employs the construction of an artificial bypass between the coronary artery and aorta (if using SV) or between the coronary artery and ITA (in the case of single or bilateral ITA grafting) and therefore links the coronary artery with a venous or arterial conduit, bringing their ECs into close proximity [24,25,26,27]. MAG, a CABG variation which employs both ITAs or the left ITA and radial artery to build a bypass, has been associated with better survival, freedom from major adverse cardiovascular events, and quality of life in comparison with SAG which relies on using ITA and SV [33,34,35,36,37,38,39,40,41,42,43,44]. Earlier, we showed beneficial effects of co-culturing HCAECs and HITAECs (i.e., elevated nitric oxide synthase expression, downregulation of endothelial-to-mesenchymal transition transcription factors Snail and Slug, and release of pro-angiogenic molecules) in comparison with culturing HCAECs together with human SV ECs (HSaVEC) [91].

Hence, we suggested that synergistic interactions between the coronary and ITA endothelium are among the factors determining better clinical outcomes of MAG, in addition to (1) similar microanatomy, as the coronary artery, ITA, and radial artery are muscular arteries which tunica media is delineated with internal and external elastic laminae and contains multiple layers of VSMCs [60,61,62,63,64,65,66,67,68,69]; (2) anti-fibrotic, anti-atherogenic, and anti-proteolytic chemical composition depleted in collagen and enriched with decorin in combination with curbed protease activity in arteries [70,71,72,73]; (3) contractile phenotype of arterial VSMCs [60,74,75,76,77,78,79,80,81,82,83]; (4) favourable secretory profile of arterial ECs, which release increased levels of vasodilators (such as nitric oxide) and pro-angiogenic molecules [84,85,86,87,88,89,90,91,92,93,94]; (5) resistance of arterial cell populations to oxidative stress, an imperative consequence of inflammation [92,93,94]; and (6) better vasoreactivity [95,96,97,98,99] and anti-thrombotic profile [100,101] of arterial ECs and VSMCs. As ECs play a key role in maintaining physiological vasodilation in response to blood pressure, have angiocrine function in terms of released extracellular vesicles and soluble factors, and ensure resistance of blood vessels to thrombosis [1,2,3,4,5], reciprocal and beneficial interactions between HCAECs and HITAECs might be of pathophysiological significance but largely depend on their heterogeneity. In this study, we aimed to perform an unbiased analysis of major proteins and global gene expression in HCAECs and HITAECs lysates by means of UHPLC-MS/MS and RNA sequencing, respectively. Rather than relying on single specific markers, we have analysed bioinformatic categories (i.e., molecular terms) encompassing the entire set of genes or proteins which are engaged in various biological processes (including biochemical pathways), share certain molecular functions, or are located in distinct cellular or extracellular compartments. Pathway enrichment analysis of molecular terms through Gene Ontology, UniProtKB Keywords, Reactome, and Kyoto Encyclopedia of Genes and Genomes (KEGG) databases included calculation of observed-verses-expected number of DEGs or DEPs in bioinformatic categories related to the structure and assembly of the endothelial monolayer (EC-ECM and EC-BM adhesion, organisation of intercellular junctions, cytoskeletal rearrangements, and synthesis and folding of ECM components) and to endothelial function (ubiquitination and autophagy, intracellular transport and extracellular secretion, bioenergetic processes, DNA repair, transcription, and translation).

We found that both HCAECs and HITAECs showed the molecular signatures of extracellular matrix (ECM) production, basement membrane (BM) synthesis, cell–cell and cell–ECM interactions, and extracellular vesicle (EV) release. Formation of BM, collagen biosynthesis and folding, and organisation of cell–cell contacts were particularly pronounced in HCAECs, whilst molecular signatures of pro-inflammatory signaling, protein and nitrogen generation, and ribosome biogenesis were overrepresented in HITAECs. Though transcriptome analysis showed notable heterogeneity between HCAECs and HITAECs, no specific markers of either EC type have been found at the protein level. As the molecular categories which were upregulated in HCAECs and HITAECs were coherent, we suggested synergistic interactions between these ECs, in agreement with our previous findings [91]. In accord to this hypothesis, the bioinformatic analysis of the molecular interactions revealed that arterial homeostasis pathways are significantly enriched in the dataset of DEPs and DEGs between HCAECs and HITAECs. Collectively, these findings provide a molecular rationale for extended use of ITA grafts and particularly for the ROMA trial, which is now conducted to evaluate the clinical arguments for prioritising MAG over SAG and to provide a definitive answer whether total arterial revascularisation and bilateral ITA grafting are beneficial at least for certain patient cohorts.

Bioinformatic analysis of proteomic and transcriptomic data obtained from three sequential experiments revealed 244 DEPs and from 729 to 1014 DEGs upregulated in HCAECs, 287 DEPs and from 849 to 926 DEGs upregulated in HITAECs, and 2794 proteins as well as from 17,789 to 19,228 genes which have not been differentially expressed between HCAECs and HITAECs (if applying a logarithmical fold change ≥ 1 and FDR-corrected *p* value ≤ 0.05). Moreover, the HCAEC/HITAEC interactome (i.e., the DEPs and DEGs having ≥1 interaction) included 254 DEPs, 516 DEGs at laminar flow, and 466 DEGs under static conditions. Among the molecular terms differentially expressed in both HCAECs and HITAECs were those accountable for the biosynthesis and folding of the ECM and BM components (e.g., collagen), adhesion of ECs to the ECM and BM, and formation of intercellular junctions. Previous studies of arterial wall composition have shown that type I, type III, type IV, type V, and type VI collagen, elastin, fibronectin, vitronectin, chondroitin sulfate, dermatan sulfate, decorin, versican, and biglycan are the major components of medial and adventitial ECM (although medial ECM is enriched and adventitial ECM is deficient in elastin), whilst the BM consists of type IV, type XV, and type XVIII collagen, laminin, nidogen, perlecan, agrin, and fibronectin [102,103,104,105,106]. Adhesion of ECs to the ECM and BM is largely mediated by integrins, whereas junctions between the ECs are formed by claudins and occludin (tight junctions constructing the intercellular barrier near the luminal surface), VE-cadherin (adherens junctions located mostly at the lateral surfaces joining adjacent ECs within the monolayer), and connexins (gap junctions passing ions and small molecules through the intercellular channels) [107,108,109,110]. Assembly of endothelial monolayer and proper regulation of endothelial integrity, geometry, and permeability ensure original specification, physiological elongation, and favourable molecular profile of arterial ECs at high shear stress [111,112,113]. In addition, chemical composition of cell membrane, which acts as a scaffold for the formation of intercellular junctions, also regulates endothelial response to high shear stress which is typical for arteries [114]. Therefore, similarity of molecular terms related to the ECM, BM, and formation of intercellular junctions in HCAECs and HITAECs suggested the structural congruence of HCAECs and HITAECs in the context of artificial coronary bypass. Intriguingly, glycosaminoglycans of the BM such as chondroitin sulfate, heparan sulfate, or dermatan sulfate also compose the glycocalyx, a meshwork of membrane-bound proteoglycans and glycoproteins lining the luminal surface of the ECs, and upregulation of these molecules in both HCAECs and HITAECs is fruitful to maintain endothelial physiology [115,116,117,118].

Further, HCAECs and HITAECs overexpressed numerous functional molecular terms including synthesis of mitochondrial ribosomes, mitochondrial transcription and translation, energy generation in mitochondria, Ca^2+^-dependent signaling, secretion of extracellular vesicles, DNA metabolism and repair, RNA processing, ubiquitination, and autophagy. Such upregulation patterns suggested rapid metabolism to support the biosynthesis of ECM and BM components indicated above, similar to the scenario occurring in tissue-engineered vascular grafts during the replacement of biodegradable polymers with newly formed vascular tissue [119,120]. Another reason for active energy consumption is the need to support and regulate endothelial barrier semi-permeability during the vasoreactivity and regulation of blood pressure. This task requires large amounts of ATP to rearrange the intercellular junctions and maintain intracellular Ca^2+^ metabolism, which is also indispensable for the enzymatic digestion and biosynthesis of nitric oxide and prostacyclin, the potent vasodilators [121,122,123,124]. With regards to nucleotide metabolism, DNA repair defects have been associated with age-related and premature endothelial dysfunction which develops as reduced nitric oxide release and chronic low-grade inflammation [125,126,127,128,129,130,131]. The same has been reported for decreased ubiquitination and impaired autophagy, as removal of dysfunctional organelles, supramolecular complexes, and molecules also becomes compromised with age [132,133,134], and ubiquitination participates in regulating rearrangements of intercellular junctions and integrity of the endothelial barrier [135,136]. Notably, both HCAECs and HITAECs were characterised by an upregulated export of extracellular vesicles which serve as shuttles for juxtacrine and paracrine protein transfer and mediate protective effects on heart [137] and blood vessels [138,139], having been proposed as potential therapeutic agent to treat myocardial ischemia [140,141,142].

Importantly, the dataset of interacting DEPs and DEGs between HCAECs and HITAECs was enriched with molecular terms related to arterial homeostasis such as those related to cell–cell and cell–matrix junctions or their positive regulation, basement membrane assembly, angiogenesis (in particular sprouting angiogenesis) and its positive regulation, proliferation and migration of ECs (as these processes are indispensable for angiogenesis), nitric oxide biosynthesis, and formation of elastic fibers. Along with coherent structural and functional molecular terms enriched in HCAECs and HITAECs proteome/transcriptome, such an in silico interactome profile suggests the synergistic impact of HCAECs and HITAECs into the functioning of the continuous endothelial monolayer within the coronary bypass in the CABG surgery context.

Although we confirmed the substantial molecular heterogeneity of HCAECs and HITAECs (244 and 287 DEPs, respectively), we did not find any definitive protein marker (having a logarithmical fold change > 5 and relevant for endothelial biology) specific for either of the mentioned ECs, and multiple molecular terms were concurrently upregulated in both of these cell lines. This suggested that differences in biochemical pathways do not affect their paracrine interactions to a major extent and pointed to their functional congruence in the context of artificial anastomosis. Likewise, the absence of specific protein markers of distinct EC lineages in adventitial vasa vasorum within the SV of patients advised for CABG surgery has been previously reported by our group [143,144]. Although the time of EC culture under laminar flow and at static conditions have not coincided and therefore these experiments could not be compared head-to-head, the culture of ECs at laminar flow is preferrable to recapitulate the physiological conditions. The comparison of transcriptomic and proteomic data demonstrated that number of DEGs (from 729 to 1014) exceeds the number of DEPs (from 244 to 287) 3- to 5-fold if applying the same logarithmical fold change (≥2), which is probably because of post-transcriptional and translational regulation and higher coverage of transcriptomic analysis. The study limitations included the inability to compare the proteomic data at laminar flow and under static conditions as the amount of total protein extracted from ECs cultured in flow chambers was insufficient to perform an adequate UHPLC-MS/MS analysis.

To conclude, we showed that the molecular profile of HCAECs and HITAECs has common upregulated molecular terms which are related to the structure of the endothelial monolayer and describe numerous aspects of EC physiology, whilst the HCAEC/HITAEC interactome is enriched with arterial homeostasis pathways. Taken together, these findings provide the arguments confirming molecular congruence of the coronary artery and ITA in the context of bypass surgery that contributes to better outcomes of MAG in comparison with SAG. Although molecular heterogeneity between HCAECs and HITAECs is undisputed and is quantified by up to ≈300 DEPs and ≈1000 DEGs, neither of these EC lines showed any distinctive protein markers.

## 4. Materials and Methods

### 4.1. Cell Culture

For the experiments, we used commercially available primary HCAECs (300K-05a, Cell Applications, San Diego, CA, USA) and HITAECs (308K-05a, Cell Applications, San Diego, CA, USA) which have been cryopreserved at the 2nd passage. Cell culture was performed as in our previous studies [91,145,146,147]. Briefly, we used MesoEndo Growth Medium (212-500, Cell Applications, San Diego, CA, USA), trypsin/ethylenediaminetetraacetic acid kit (090K, Cell Applications, San Diego, CA, USA), and T-75 flasks (90076, Techno Plastic Products, Trasadingen, Switzerland) for subculturing both types of ECs. Passaging was carried out at 80–90% confluence. After 3 passages (i.e., at the 5th passage in total), HCAECs and HITAECs were seeded into flow channel slides (80126, Ibidi, Gräfelfing, Germany) or in fresh T-75 (90076, Techno Plastic Products, Trasadingen, Switzerland) for culturing cells under flow or under static conditions, respectively. In flow culture studies, HCAECs and HITAECs were seeded into flow channel slides (80126, Ibidi, Gräfelfing, Germany) at 90% confluence (≈350,000 cells per slide), left for 16 h for proper adhesion, and then cultured under laminar flow (15 dyn/cm^2^) using the respective perfusion set of 50 cm length, 1.6 mm internal diameter, and 10 mL reservoirs (Yellow/Green, 10964, Ibidi, Gräfelfing, Germany) in the flow culture system (Ibidi Pump System Quad, Ibidi, Gräfelfing, Germany) for 48 h. In static conditions, HCAECs and HITAECs were also incubated for 48 h after reaching >95% confluence (i.e., after an endothelial monolayer was established). Standard cell culture conditions (37 °C, 95% air: 5% CO_2_ atmosphere, high humidity, and sterile conditions) were continuously maintained (MCO-18AIC, Sanyo, Tokyo, Japan).

### 4.2. Whole-Transcriptome Analysis

Whole-transcriptome analysis was conducted employing RNA sequencing, which was carried out in SB RAS Genomics Core Facility (Institute of Chemical Biology and Fundamental Medicine of the Siberian Branch of the Russian Academy of Sciences, Novosibirsk). Upon the withdrawal of culture medium and washing in ice-cold phosphate buffered saline (PBS, pH 7.4, 10010023, Thermo Fisher Scientific, Waltham, MA, USA), cells (one T-75 flask (equal to ≈ 3,000,000 cells), or 4 flow channel slides (equal to ≈ 400,000 cells), n = 3 flasks or 3 four-slide batches of flow channel slides per cell line) were lysed with TRIzol (15596018, Thermo Fisher Scientific, Waltham, MA, USA) with the following total RNA isolation (Purelink RNA Micro Scale Kit, 12183016, Thermo Fisher Scientific, Waltham, MA, USA) and DNAse treatment (DNASE70, Sigma-Aldrich, Saint Louis, MO, USA) according to the manufacturer’s protocols. The RNA integrity index (RIN) was assessed using an RNA 6000 Pico Kit (5067-1513, Agilent Technologies, Santa Clara, CA, USA) and Bioanalyzer 2100 (Agilent Technologies, Santa Clara, CA, USA). RNA quantification was carried out using NanoDrop 2000 (Thermo Fisher Scientific, Waltham, MA, USA) and Qubit 4 (Thermo Fisher Scientific, Waltham, MA, USA).

For the 1 µg of isolated RNA, we performed rRNA depletion (RiboCop rRNA Depletion Kit V1.2, 037.96, Lexogen, Wien, Austria) followed by DNA library preparation (MGIEasy RNA Directional Library Prep Set, MGI Tech, Shenzhen, China) and quality control (High Sensitivity DNA Kit, 5067-4626, Agilent Technologies, Santa Clara, CA, USA). DNA libraries were then quantified by qPCR (CFX96 Touch, Bio-Rad, Hercules, CA, USA), pooled in equimolar amounts, and sequenced (MGIseq-2000, MGI Tech, Shenzhen, China) using a DNBSEQ-G400RS High-throughput Sequencing Set (FCL PE100, 1000016950, MGI Tech, Shenzhen, China).

Read mapping to the human genome (hg38) was conducted using STAR v2.7.6a [148]. Gene expression (exon coverage) was evaluated by HTSeq-count v0.12.4 [149] using Ensembl annotation (v.38.93). To identify DEGs, we used DESeq2 package (https://bioconductor.org/packages/release/bioc/html/DESeq2.html (accessed on 25 August 2023)) [150]. DEGs were defined as those with a logarithmical fold change ≥ 1 and false discovery rate (FDR)-corrected *p* value ≤ 0.05. Bioinformatic analysis was performed using Gene Ontology [151,152], UniProtKB Keywords [153], Reactome [154,155], and Kyoto Encyclopedia of Genes and Genomes (KEGG) [156,157] databases. The RNA sequencing data have been deposited in Sequence Read Archive (accession number: PRJNA891895), “https://www.ncbi.nlm.nih.gov/bioproject/PRJNA891895/” (accessed on 25 August 2023).

### 4.3. Proteomic Profiling

Proteomic profiling was performed by means of UHPLC-MS/MS with ion mobility in the Centre for Molecular and Cell Technologies (Saint Petersburg State University Research Park). Upon the withdrawal of culture medium and washing in ice-cold phosphate buffered saline (PBS, pH 7.4, 10010023, Thermo Fisher Scientific, Waltham, MA, USA), HCAECs and HITAECs (one T-75 flask, equal to ≈ 3,000,000 cells, *n* = 3 flasks) were lysed with a radioimmunoprecipitation assay (RIPA) buffer (89901, Thermo Fisher Scientific, Waltham, MA, USA) supplied with a Halt protease and phosphatase inhibitor cocktail (78444, Thermo Fisher Scientific, Waltham, MA, USA) according to the manufacturer’s protocol and centrifuged at 14,000× *g* (Microfuge 20R, Beckman Coulter, Brea, CA, USA) for 15 min. Protein quantification was conducted using a BCA Protein Assay Kit (23227, Thermo Fisher Scientific, Waltham, MA, USA) and Multiskan Sky microplate spectrophotometer (Thermo Fisher Scientific, Waltham, MA, USA) in accordance with the manufacturer’s protocol.

To prepare the samples for the tryptic digestion, we first removed the RIPA buffer by 1-h acetone precipitation (650501, Sigma-Aldrich, Saint Louis, MO, USA) at –20 °C and centrifugation at 13,000× *g* for 15 min at 4 °C (Microfuge 20R, Beckman Coulter, Brea, CA, USA). Then, the protein pellet was resuspended in 250 µL acetone for 15 min at −20 °C and centrifugation was repeated. After removal of the supernatant and evaporating the residual acetone for 5–10 min, the protein pellet was resuspended in 8 mol/L urea (U5128, Sigma-Aldrich, Saint Louis, MO, USA) in 50 mmol/L ammonium bicarbonate (09830, Sigma-Aldrich, Saint Louis, MO, USA), incubated for 20 min on ice (at 4 °C), ultrasonicated in a water bath (Sapphire, Moscow, Russia), and incubated for another 10 min on ice (at 4 °C). The protein concentration was measured by a Qubit 4 fluorometer (Q33238, Thermo Fisher Scientific, Waltham, MA, USA) with a QuDye Protein Quantification Kit (25102, Lumiprobe, Cockeysville, MD, USA) according to the manufacturer’s protocol. Protein samples (15 μg) were then incubated in 5 mmol/L dithiothreitol (D0632, Sigma-Aldrich, Saint Louis, MO, USA) for 1 h at 37 °C with the subsequent incubation in 15 mmol/L iodoacetamide for 30 min in the dark at room temperature (I1149, Sigma-Aldrich, Saint Louis, MO, USA). Next, the samples were diluted with 7 volumes of 50 mmol/L ammonium bicarbonate and incubated for 16 h at 37 °C with 300 ng of trypsin (1:50 trypsin:protein ratio; VA9000, Promega, Madison, WI, USA). The peptides were then desalted with stage tips (Tips-RPS-M.T2.200.96, Affinisep, Le Houlme, France) according to the manufacturer’s protocol using methanol (1880092500, Sigma-Aldrich, Saint Louis, MO, USA), acetonitrile (1000291000, Sigma-Aldrich, Saint Louis, MO, USA), and 0.1% formic acid (33015, Sigma-Aldrich, Saint Louis, MO, USA). Desalted peptides were dried in a centrifuge concentrator (Concentrator plus, Eppendorf, Hamburg, Germany) for 3 h and finally dissolved in water for chromatography (1153334000, Sigma-Aldrich, Saint Louis, MO, USA) supplied with 0.1% formic acid (33015, Sigma-Aldrich, Saint Louis, MO, USA) for the further shotgun proteomics analysis.

Approximately 500 ng of peptides per sample were used for shotgun proteomics analysis by UHPLC-MS/MS with ion mobility in a trapped ion mobility spectrometry time-of-flight (TimsToF) Pro mass spectrometer with a nanoElute UHPLC system (Bruker, Billerica, MA, USA). UHPLC was performed in the two-column separation mode with an Acclaim PepMap 5 mm Trap Cartridge (Thermo Fisher Scientific) and a Bruker Fifteen separation column (C18 ReproSil AQ, 150 mm × 0.75 mm, 1.9 µm, 120 A; Bruker, Billerica, MA, USA) in a gradient mode with 400 nL/min flow rate and 40 °C. Phase A was water/0.1% formic acid (33015, Sigma-Aldrich, Saint Louis, MO, USA) and phase B was acetonitrile/0.1% formic acid (1000291000, Sigma-Aldrich, Saint Louis, MO, USA; 33015, Sigma-Aldrich, Saint Louis, MO, USA). The gradient was from 2% to 30% phase B for 42 min, then to 95% phase B for 6 min with subsequent washing with 95% phase B for 6 min. Before each sample, trap and separation columns were equilibrated with 10 and 4 column volumes, respectively.

CaptiveSpray ion source was used for electrospray ionization with 1600 V of capillary voltage, 3 L/min N_2_ flow, and 180 °C source temperature. The mass spectrometry acquisition was performed in data-dependent acquisition parallel accumulation serial fragmentation (DDA-PASEF) mode with 0.5 s cycle in positive polarity with the fragmentation of ions with at least two charges in an m/z range from 100 to 1700 and ion mobility range from 0.85 to 1.30 1/K0. Protein identification was performed in PEAKS Studio Xpro software (a license granted to St. Petersburg State University; Bioinformatics Solutions, Waterloo, ON, Canada) “https://www.bioinfor.com/peaks-software/” (accessed on 25 August 2023) using the human protein SwissProt database “https://www.uniprot.org/” (accessed on 20 July 2022; organism: Human [9606]; uploaded on 2 March 2021; 20,394 sequences) and protein contaminants database CRAP (version of 4 March 2019). The search parameters were as follows: parent mass error tolerance 10 ppm and fragment mass error tolerance 0.05 ppm, protein and peptide FDR <1% and 0.1%, respectively, two possible missed cleavage sites, and proteins with ≥2 unique peptides. Cysteine carbamidomethylation was set as fixed modification. Methionine oxidation, N-terminal acetylation, and asparagine and glutamine deamidation were set as variable modifications.

Then, label-free quantification by peak area under the curve was used for further analysis in R (version 3.6.1; R Core Team, 2019). The biological groups were qualitatively compared by the “VennDiagram” package “https://cran.r-project.org/web/packages/VennDiagram/index.html” (accessed on 25 August 2023) [158]. Then, proteins with missing values in >2 replicates were removed, and the rest of the missing values were imputed by the k-nearest neighbours algorithm. Multivariate data analysis for quantitative protein expression data was performed by non-metric multidimensional scaling using the “Vegan” package “https://cran.r-project.org/web/packages/vegan/index.html” (accessed on 25 August 2023) [159] and by clusterisation of samples by sparse partial least squares discriminant analysis in the “MixOmics” package “https://bioconductor.org/packages/release/bioc/html/mixOmics.html” (accessed on 25 August 2023) [160]. Differentially expressed proteins were identified through the moderated *t*-test by the “limma” package “https://bioconductor.org/packages/release/bioc/html/limma.html” (accessed on 25 August 2023) [161] and were defined as those with a logarithmical fold change ≥1 and FDR-corrected *p* value ≤ 0.05. Pathway enrichment analysis of differentially expressed proteins was performed using Gene Ontology [151,152], UniProtKB Keywords [153], Reactome [154,155], and Kyoto Encyclopedia of Genes and Genomes (KEGG) [156,157] databases. The “ggplot2” “https://cran.r-project.org/web/packages/ggplot2/index.html” (accessed on 25 August 2023) [162] and “EnhancedVolcano” “https://bioconductor.org/packages/release/bioc/html/EnhancedVolcano.html” (accessed on 25 August 2023) [163] packages were used for visualisation. The mass spectrometry proteomics data have been deposited to the ProteomeXchange Consortium via the PRIDE [164] partner repository with the dataset identifier PXD037861.

### 4.4. Bioinformatic Analysis of the HCAEC/HITAEC Interactome

To analyse the profile of interacting DEPs and DEGs in HCAECs and HITAECs, we applied the Cytoscape software (version 3.10.1, Cytoscape Consortium, USA) [165] with a stringApp utilising the following workflow: (1) filtration of DEPs and DEGs having ≥1 interaction; (2) colour mapping in order to distinguish interacting DEPs and DEGs between HCAECs and HITAECs; (3) pathway enrichment analysis using GO, Reactome, UniProtKB, and KEGG terms; and (4) selection of such molecular terms relevant for arterial homeostasis.

### 4.5. RT-qPCR Analysis

Upon the RNA extraction for whole-transcriptome analysis, we performed reverse transcription using a High-Capacity cDNA Reverse Transcription Kit (4368814, Thermo Fisher Scientific, Waltham, MA, USA) and then measured the gene expression level by RT-qPCR. The same RNA aliquots have been used to allocate RNA for RNA-seq and RT-qPCR in order to ensure the correct verification of whole-transcriptome analysis. RT-qPCR was conducted using customised primers (500 nmol/L each, Evrogen, Moscow, Russia, Appendix A), cDNA (20 ng), and PowerUp SYBR Green Master Mix (A25778, Thermo Fisher Scientific, Waltham, MA, USA) according to the manufacturer’s protocol for Tm ≥ 60 °C (fast cycling mode). Technical replicates (n = 3 per each sample) were performed in all qPCR experiments. Quantification of the mRNA levels was performed by calculation of ΔCt and by using the 2^−ΔΔCt^ method. Relative transcript levels were expressed as a value relative to the average of three housekeeping genes (*GAPDH*, *ACTB*, and *B2M*) and to the HCAECs group (2^−ΔΔCt^).

### 4.6. Western Blotting

Equal amounts of the same protein lysates that were selected for proteomic profiling (10 μg per sample) were prepared for fluorescent Western blotting as previously described [120]. A Chameleon Duo Pre-Stained Protein Ladder (928–60000, LI-COR Biosciences, Lincoln, NE, USA) was loaded as a molecular weight marker. The proteins were separated, and the protein transfer was performed using nitrocellulose transfer stacks as previously described [120]. Nitrocellulose membranes were then incubated in protein-free Block’n’Boost! solution (K-028, Molecular Wings, Kemerovo, Russia) for 1 h to prevent non-specific binding.

The blots were probed with (1) rabbit antibodies to VCAM1 (1:250 dilution, SL0920R, Sunlong Biotech, Hangzhou, China) and mouse antibodies to GAPDH (1:1000, SLM33033M, Sunlong Biotech, Hangzhou, China); (2) rabbit antibodies to ICAM1 (1:250, SL0608R, Sunlong Biotech, Hangzhou, China) and mouse antibodies to CD31 (1:2000, ab9498, Abcam, Cambridge, UK); (3) rabbit antibodies to Snail and Slug (1:250, ab180714, Abcam, Cambridge, UK) and mouse antibodies to CD31 (1:2000, ab9498, Abcam, Cambridge, UK); (4) rabbit antibodies to ZEB1 (70512, Cell Signaling Technology, Danvers, MA, USA) and mouse antibodies to TWIST1 (1:100, sc-81417, Santa Cruz Biotechnology, Dallas, TX, USA); (5) rabbit antibodies to KLF2 (1:200, SL2772R, Sunlong Biotech, Hangzhou, China) and mouse antibodies to TWIST1 (1:200, sc-81417, Santa Cruz Biotechnology, Dallas, TX, USA); (6) rabbit antibodies to KLF4 (1:500, ab215036, Abcam, Cambridge, UK) and mouse antibodies to eNOS (1:500, SLM33176M, Sunlong Biotech, Hangzhou, China); and (7) rabbit antibodies to NRF2 (1:200, ab62352, Abcam, Cambridge, UK) and mouse antibodies to GAPDH (1:1000, SLM33033M, Sunlong Biotech, Hangzhou, China). IRDye 680RD-conjugated goat anti-mouse (926-68070, LI-COR Biosciences, Lincoln, NE, USA) and IRDye 800CW-conjugated goat anti-rabbit (926-32211, LI-COR Biosciences, Lincoln, NE, USA) or IRDye 680RD-conjugated goat anti-rabbit (926-68071, LI-COR Biosciences, Lincoln, NE, USA) and IRDye 800CW-conjugated goat anti-mouse (926-32210, LI-COR Biosciences, Lincoln, NE, USA) IgG secondary antibodies were used at a 1:1000 dilution.

Incubation with the antibodies and fluorescent detection was performed using an Odyssey XF imaging system (LI-COR Biosciences, Lincoln, NE, USA) at a 700 nm channel (685 nm excitation and 730 nm emission). Total protein normalisation was conducted after the fluorescent detection using 0.1% Fast Green FCF (F8130, Solarbio Life Sciences, Beijing, China) as previously described [120]. The total protein visualisation was performed using an Odyssey XF imaging system (LI-COR Biosciences, Lincoln, NE, USA) in a 600 nm channel (520 nm excitation and 600 nm emission).

### 4.7. Analysis of Bypass Dysfunctions after CABG Surgery at 10 Years of Follow-Up

To analyse the long-term efficiency of SV verses ITA grafts applied for creating a bypass during CABG surgery, we collected the data from 102 case histories of patients who underwent CABG and conducted a repeated coronary angiography (GE Healthcare Innova 3100 Cardiac Angiography System, General Electric Healthcare, Chicago, IL, USA) at 10 years of follow-up. The study was conducted according to the latest revision of the Declaration of Helsinki (2013), and the study protocol was approved by the Local Ethical Committee of the Research Institute for Complex Issues of Cardiovascular Diseases (Kemerovo, Russia, protocol code 021/2023, date of approval: 2 February 2023). Written informed consent has been provided by all study participants after receiving a full explanation of the study’s purposes. In total, we analysed 311 bypasses between (1) the SV and right coronary artery, obtuse marginal artery, or diagonal branches of the left anterior descending artery; (2) the left ITA and left anterior descending artery, obtuse marginal artery, or diagonal branches of the left anterior descending artery; and (3) the right ITA and right coronary artery, obtuse marginal artery, or left anterior descending artery. Adverse outcomes included development of atherosclerosis in the coronary artery, competitive flow, and graft degeneration at 10 years of follow-up. Favourable outcomes included freedom from adverse outcomes at 10 years of follow-up. Comparison of the proportions was carried out using a Pearson’s chi-squared test with Yates’s correction for continuity.

## 5. Conclusions

Bioinformatic analysis of RNA sequencing and proteomic profiling data showed that the molecular heterogeneity of HCAECs and HITAECs does not contradict their functional congruence. HCAECs and HITAECs had similar upregulated molecular terms related to their structure (adhesion to ECM and BM, biosynthesis and folding of ECM and BM, and assembly of intercellular junctions) and function (biogenesis of mitochondrial ribosomes, mitochondrial transcription and translation, energy generation in mitochondria, Ca^2+^-dependent signaling, secretion of extracellular vesicles, DNA metabolism and repair, and RNA processing and autophagy), and no specific protein markers have been revealed for either of the mentioned EC lines. This can be considered as a molecular basis of their synergistic effects which have been revealed earlier in a co-culture model. The coherence of HCAECs and HITAECs molecular profiles might also better explain the mid- and long-term outcomes of MAG, a CABG surgery modality including the use of >1 artery to create a bypass.

## Figures and Tables

**Figure 1 ijms-24-15032-f001:**
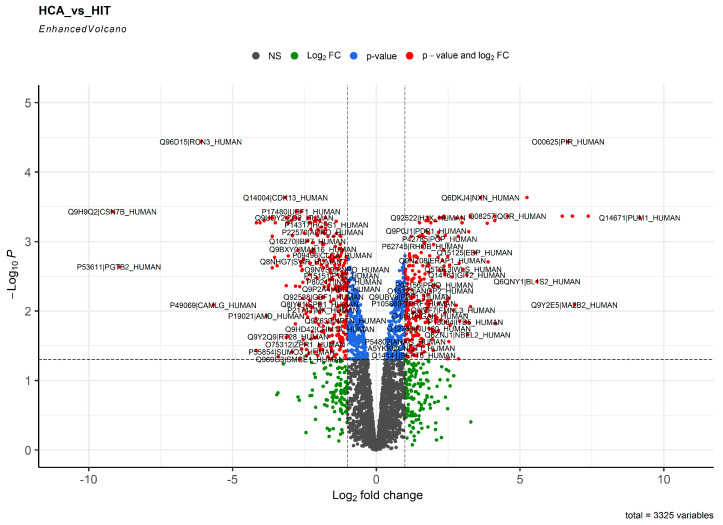
Volcano plot showing the distribution of proteins in the proteome of human coronary artery endothelial cells (HCAECs) and human internal thoracic artery endothelial cells (HITAECs). Gray points depict the proteins with log_2_ fold change < 1 and FDR-corrected *p* value > 0.05. Green points depict the proteins with log_2_ fold change > 1 and FDR-corrected *p* value > 0.05. Blue points depict the proteins with log_2_ fold change < 1 and FDR-corrected *p* value < 0.05. Red points depict the proteins with log_2_ fold change > 1 and FDR-corrected *p* value < 0.05 (DEPs).

**Figure 2 ijms-24-15032-f002:**
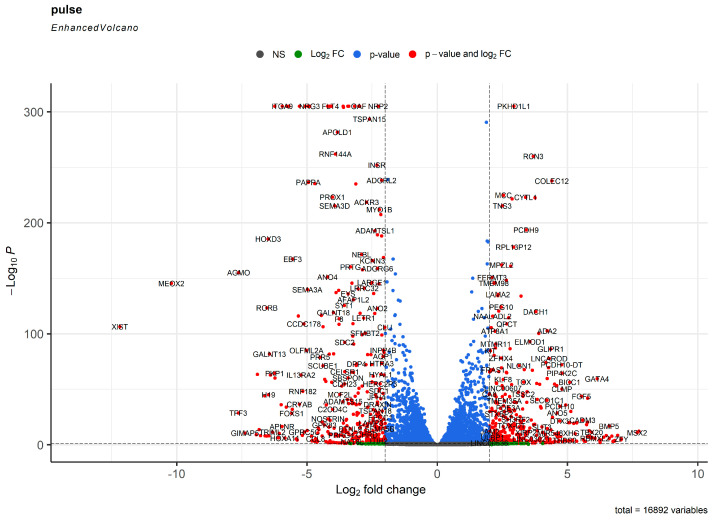
Volcano plot showing the distribution of transcripts in the transcriptome of human coronary artery endothelial cells (HCAECs) and human internal thoracic artery endothelial cells (HITAECs) under laminar flow. Gray points depict the genes with log_2_ fold change < 1 and FDR-corrected *p* value > 0.05. Green points depict the genes with log_2_ fold change > 1 and FDR-corrected *p* value > 0.05. Blue points depict the genes with log_2_ fold change < 1 and FDR-corrected *p* value < 0.05. Red points depict the genes with log_2_ fold change > 1 and FDR-corrected *p* value < 0.05 (DEGs).

**Figure 3 ijms-24-15032-f003:**
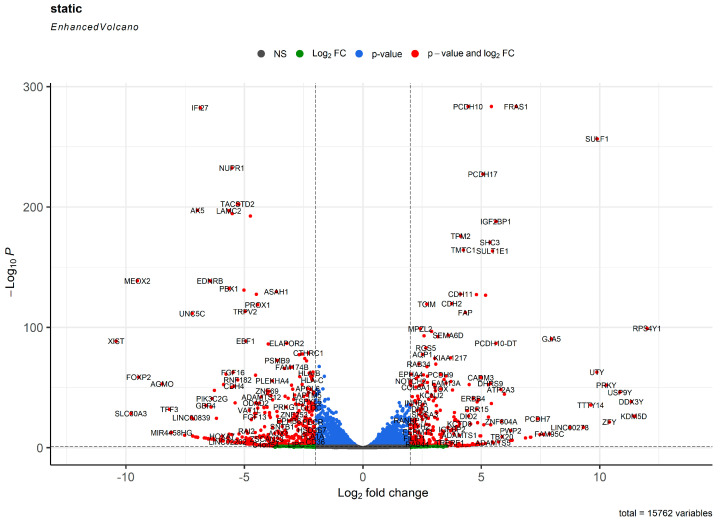
Volcano plot showing the distribution of transcripts in the transcriptome of human coronary artery endothelial cells (HCAECs) and human internal thoracic artery endothelial cells (HITAECs) at static cell culture conditions. Gray points depict the genes with log_2_ fold change < 1 and FDR-corrected *p* value > 0.05. Green points depict the genes with log_2_ fold change > 1 and FDR-corrected *p* value > 0.05. Blue points depict the genes with log_2_ fold change < 1 and FDR-corrected *p* value < 0.05. Red points depict the genes with log_2_ fold change > 1 and FDR-corrected *p* value < 0.05 (DEGs).

**Figure 4 ijms-24-15032-f004:**
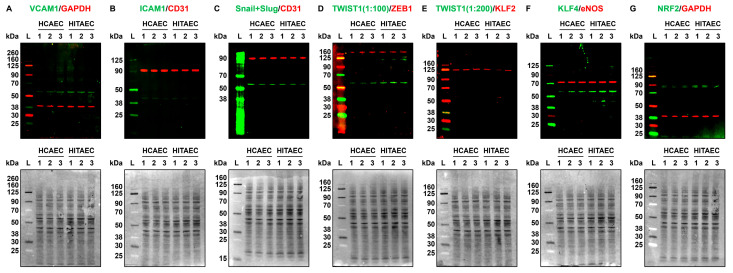
Fluorescent Western blotting for cell adhesion molecules (VCAM1 and ICAM1), transcription factors of endothelial-to-mesenchymal transition (Snail and Slug, TWIST1, and ZEB1), mechanosensitive transcription factors (KLF2, KLF4, and NRF2), endothelial nitric oxide synthase eNOS, and loading control (GAPDH and CD31) in human coronary artery endothelial cells (HCAECs) and human internal thoracic artery endothelial cells (HITAECs) cultured at static conditions. (**A**) VCAM1 (pro-inflammatory cell adhesion molecule, green)/GAPDH (loading control, red), fluorescent Western blot (**top**), and total protein staining confirming an equal protein loading (**bottom**); (**B**) ICAM1 (pro-inflammatory cell adhesion molecule, green)/CD31 (loading control, red), fluorescent Western blot (**top**), and total protein staining confirming an equal protein loading (**bottom**); (**C**) Snail and Slug (endothelial-to-mesenchymal transition transcription factor, green)/CD31 (loading control, red), fluorescent Western blot (**top**), and total protein staining confirming an equal protein loading (**bottom**); (**D**) TWIST1 (endothelial-to-mesenchymal transition transcription factor, green)/ZEB1 (another endothelial-to-mesenchymal transition transcription factor, red), fluorescent Western blot (**top**), and total protein staining confirming an equal protein loading (**bottom**); (**E**) TWIST1 (endothelial-to-mesenchymal transition transcription factor, green)/KLF2 (atheroprotective mechanosensitive transcription factor, red), fluorescent Western blot (**top**), and total protein staining confirming an equal protein loading (**bottom**); (**F**) KLF4 (atheroprotective mechanosensitive transcription factor, green)/eNOS (endothelial nitric oxide synthase, red), fluorescent Western blot (**top**), and total protein staining confirming an equal protein loading (**bottom**); (**G**) NRF2 (atheroprotective mechanosensitive transcription factor, green)/GAPDH (loading control, red), fluorescent Western blot (**top**), and total protein staining confirming an equal protein loading (**bottom**). Each band within the groups represent a protein lysate from one experiment (*n* = 3 experiments in total). Total protein normalisation was conducted by Fast Green FCF staining of the membranes after the fluorescent imaging to ensure the equal protein loading at all blots (in addition to loading controls such as GAPDH or CD31). Fluorescent ladder (L) and molecular weight signatures (kDa) are provided to the left of the HCAECs and HITAECs protein bands. Ratios of 1:100 and 1:200 are dilutions of the antibody against TWIST1, highlighted to show low expression of this protein in the quiescent ECs.

**Figure 5 ijms-24-15032-f005:**
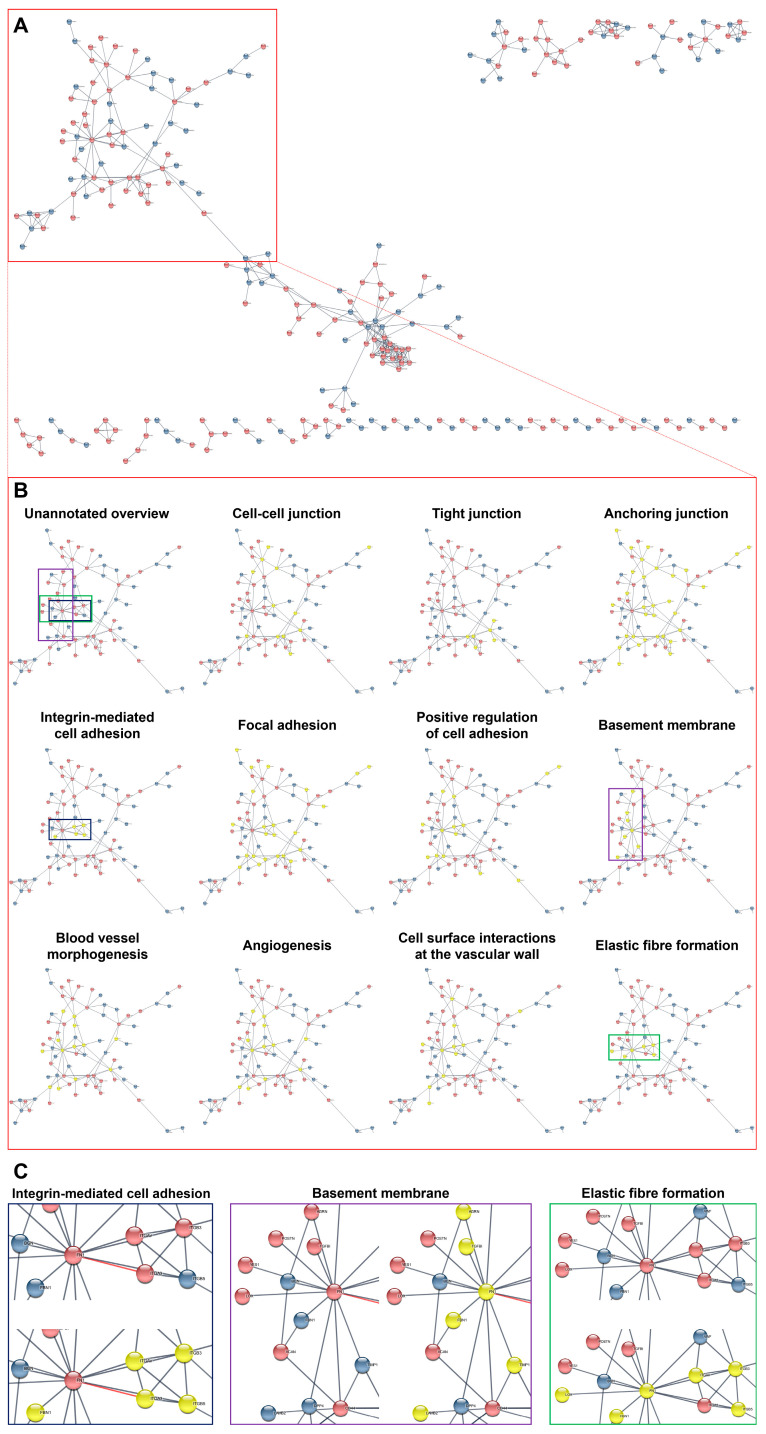
Bioinformatic analysis of protein–protein interactions between primary human coronary endothelial cells (HCAECs) and human internal thoracic artery endothelial cells (HITAECs). (**A**) Overview of protein–protein interactions between HCAECs (blue colour) and HITAECs (red colour); (**B**) annotation of the interacting proteins related to the specific molecular terms (yellow colour) in comparison with unannotated overview (the main cluster of interacting proteins is demarcated by red contour); and (**C**) annotations of clustered interacting proteins in selected molecular terms: integrin-mediated cell adhesion (demarcated by blue contour), basement membrane (demarcated by violet contour), and elastic fiber formation (demarcated by green contour).

**Figure 6 ijms-24-15032-f006:**
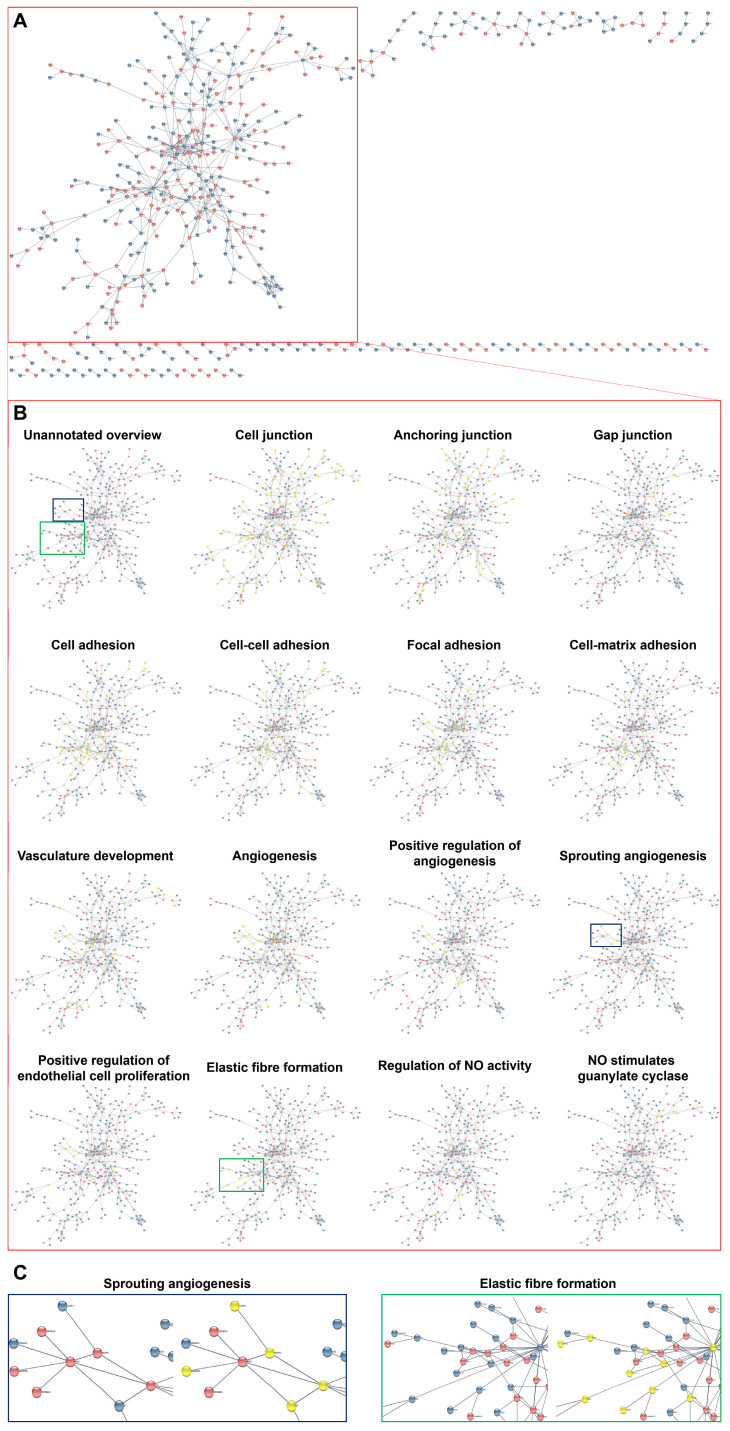
Bioinformatic analysis of gene–gene interactions between primary human coronary endothelial cells (HCAECs) and human internal thoracic artery endothelial cells (HITAECs) cultured under laminar flow. (**A**) Overview of gene–gene interactions between HCAECs (blue colour) and HITAECs (red colour) cultured under laminar flow; (**B**) annotation of the interacting genes related to the specific molecular terms (yellow colour) in comparison with unannotated overview (the main cluster of interacting genes is demarcated by red contour); and (**C**) annotations of clustered interacting genes in selected molecular terms: sprouting angiogenesis (demarcated by blue contour) and elastic fiber formation (demarcated by green contour).

**Figure 7 ijms-24-15032-f007:**
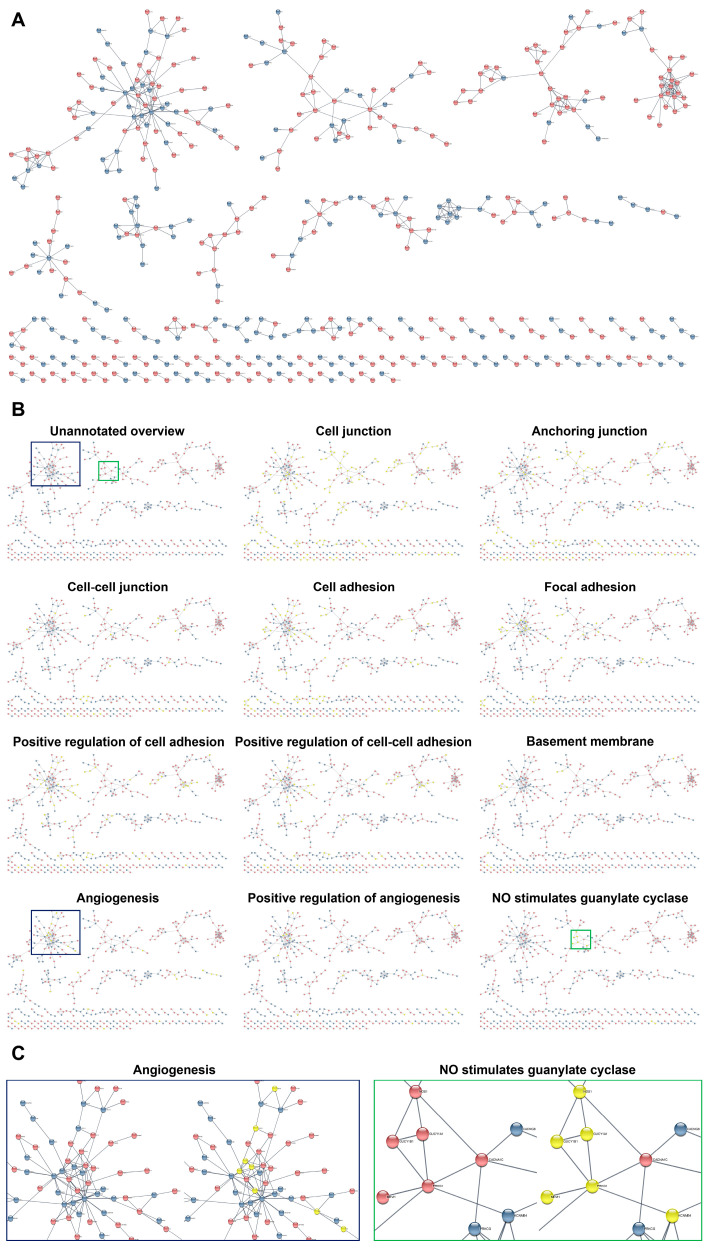
Bioinformatic analysis of gene–gene interactions between primary human coronary endothelial cells (HCAECs) and human internal thoracic artery endothelial cells (HITAECs) cultured at static conditions. (**A**) Overview of gene–gene interactions between HCAECs (blue colour) and HITAECs (red colour) cultured under laminar flow; (**B**) annotation of the interacting genes related to the specific molecular terms (yellow colour) in comparison with unannotated overview; and (**C**) annotations of clustered interacting genes in selected molecular terms: angiogenesis (demarcated by blue contour) and stimulation of guanylate cyclase by nitric oxide (demarcated by green contour).

**Table 1 ijms-24-15032-t001:** Reverse transcription-quantitative polymerase chain reaction (RT-qPCR) analysis of the gene expression in human coronary artery endothelial cells (HCAECs) and human internal thoracic artery endothelial cells (HITAECs) cultured at laminar flow or in static conditions. Expression has been normalised for the expression of three housekeeping genes (*GAPDH*, *ACTB*, and *B2M*).

Gene	Flow Culture Chambers (µ-Slide y-Shaped)			Static Conditions Experiment #1			Static Conditions Experiment #2			Static Conditions Experiment #3			Static Conditions Experiment #4		
	HCAECs, ΔCt (Mean ± Standard Deviation)	HITAECs, ΔCt (Mean ± Standard Deviation)	Fold Change (HCAECs to HITAECs)	HCAECs, ΔCt (Mean ± Standard Deviation)	HITAECs, ΔCt (Mean ± Standard Deviation)	Fold Change (HCAECs to HITAECs)	HCAECs, ΔCt (Mean ± Standard Deviation)	HITAECs, ΔCt (Mean ± Standard Deviation)	Fold Change (HCAECs to HITAECs)	HCAECs, ΔCt (Mean ± Standard Deviation)	HITAECs, ΔCt (Mean ± Standard Deviation)	Fold Change (HCAECs to HITAECs)	HCAECs, ΔCt (Mean ± Standard Deviation)	HITAECs, ΔCt (Mean ± Standard Deviation)	Fold Change (HCAECs to HITAECs)
*VCAM1*	0.0051 ± 0.0004	0.0053 ± 0.0006	0.962	0.0015 ± 0.0004	0.0098 ± 0.0009	0.153	N/D	N/D	N/D	0.0013 ± 0.0002	0.0955 ± 0.0039	0.014	0.0051 ± 0.0002	0.0455 ± 0.0041	0.112
*ICAM1*	0.0331 ± 0.0021	0.1659 ± 0.0121	0.200	0.0050 ± 0.0003	0.0407 ± 0.0027	0.123	0.0279 ± 0.0132	0.0559 ± 0.0231	0.499	0.0163 ± 0.0079	2.0781 ± 0.7478	0.008	0.0313 ± 0.0096	0.2675 ± 0.1376	0.117
*SELE*	0.0012 ± 0.0001	0.0265 ± 0.0067	0.045	0.0011 ± 0.0002	0.0481 ± 0.0039	0.023	0.0002 ± 0.00002	0.0219 ± 0.0016	0.009	0.0004 ± 0.0001	0.0942 ± 0.0110	0.004	0.0029 ± 0.0002	0.0939 ± 0.0057	0.031
*SELP*	0.0010 ± 0.0001	0.0184 ± 0.0028	0.054	0.0025 ± 0.0009	0.0044 ± 0.0011	0.568	0.0899 ± 0.0035	0.0080 ± 0.0037	11.238	0.0127 ± 0.0016	0.0555 ± 0.0099	0.229	0.0321 ± 0.0022	0.0203 ± 0.0011	1.581
*IL6*	0.0044 ± 0.0004	0.0199 ± 0.0016	0.221	0.0029 ± 0.0007	0.0007 ± 0.0003	4.143	0.0042 ± 0.0003	0.0211 ± 0.0019	0.199	0.0029 ± 0.0002	0.0095 ± 0.0020	0.305	0.0084 ± 0.0009	0.0050 ± 0.0003	1.680
*CXCL8*	0.0337 ± 0.0009	0.1774 ± 0.0129	0.190	0.0701 ± 0.0045	0.0557 ± 0.0051	1.259	0.0256 ± 0.0029	0.3286 ± 0.0285	0.078	0.0301 ± 0.0028	0.5886 ± 0.0732	0.051	0.1107 ± 0.0063	0.4469 ± 0.0335	0.248
*CCL2*	0.0112 ± 0.0009	0.2993 ± 0.0247	0.037	0.0279 ± 0.0091	0.2563 ± 0.0137	0.109	0.0614 ± 0.0038	1.0850 ± 0.1059	0.057	0.0543 ± 0.0026	4.3406 ± 0.9546	0.013	0.0920 ± 0.0048	0.9923 ± 0.0732	0.093
*CXCL1*	N/D	N/D	N/D	0.0763 ± 0.0080	0.0651 ± 0.0061	1.172	0.0984 ± 0.0274	0.2571 ± 0.0556	0.383	0.1074 ± 0.0154	1.6440 ± 0.4153	0.065	0.2617 ± 0.0185	0.2715 ± 0.0439	0.964
*MIF*	N/D	N/D	N/D	0.2152 ± 0.0231	0.0812 ± 0.0074	2.650	0.4353 ± 0.0956	2.6439 ± 0.1977	0.165	0.1903 ± 0.0117	4.0451 ± 0.5167	0.047	0.4234 ± 0.0139	0.6183 ± 0.0129	0.685
*KLF2*	0.0047 ± 0.0006	0.0021 ± 0.0002	2.238	0.0018 ± 0.0009	0.0004 ± 0.0001	4.500	N/D	N/D	N/D	N/D	N/D	N/D	N/D	N/D	N/D
*KLF4*	0.0239 ± 0.0012	0.1268 ± 0.0178	0.188	0.0010 ± 0.0002	0.0005 ± 0.0001	2.000	N/D	N/D	N/D	N/D	N/D	N/D	N/D	N/D	N/D
*NFE2L2*	0.0618 ± 0.0041	0.0293 ± 0.0010	2.109	0.0866 ± 0.088	0.0523 ± 0.0026	1.656	N/D	N/D	N/D	N/D	N/D	N/D	N/D	N/D	N/D
*SNAI1*	0.0312 ± 0.0011	0.3193 ± 0.0632	0.098	0.0071 ± 0.0004	0.0121 ± 0.0017	0.587	0.0286 ± 0.0031	0.0381 ± 0.0065	0.751	0.0011 ± 0.0001	0.1209 ± 0.0140	0.009	0.0236 ± 0.0011	0.0237 ± 0.0031	0.996
*SNAI2*	0.0003 ± 0.00002	0.0002 ± 0.00004	1.500	0.0002 ± 0.00001	0.0001 ± 0.00001	2.000	0.0371 ± 0.0045	0.0025 ± 0.0007	14.840	0.00006 ± 0.00002	0.0043 ± 0.0015	0.014	0.0034 ± 0.0002	0.0001 ± 0.00005	34.000
*TWIST1*	0.0005 ± 0.0001	0.0001 ± 0.00001	5.000	0.0004 ± 0.00001	0.00001 ± 0.00001	40.000	0.0010 ± 0.0004	0.0007 ± 0.0002	1.429	0.00008 ± 0.00002	0.00014 ± 0.0001	0.571	0.0002 ± 0.00004	0.00004 ± 0.00002	5.000
*ZEB1*	0.0845 ± 0.0096	0.0886 ± 0.0056	0.954	0.0350 ± 0.0057	0.0343 ± 0.0034	1.020	0.0818 ± 0.0037	0.1628 ± 0.0196	0.502	0.0068 ± 0.0004	0.0655 ± 0.0094	0.104	0.0486 ± 0.0024	0.0552 ± 0.0083	0.880
*CDH5*	0.6792 ± 0.0132	3.0981 ± 0.3207	0.219	0.4247 ± 0.0985	0.5115 ± 0.0189	0.830	0.3888 ± 0.0087	0.3852 ± 0.0165	1.009	0.9632 ± 0.0871	9.4160 ± 2.8673	0.102	1.1429 ± 0.1359	2.1643 ± 0.0577	0.528
*CDH2*	0.0088 ± 0.0013	0.0103 ± 0.0015	0.854	0.0108 ± 0.0015	0.0028 ± 0.0009	3.857	0.0670 ± 0.0056	0.0001 ± 0.0002	670.000	0.0479 ± 0.0032	0.0146 ± 0.0039	3.281	0.1393 ± 0.0119	0.0079 ± 0.0003	17.633
*NOS3*	0.0450 ± 0.0008	0.1536 ± 0.0102	0.293	0.0066 ± 0.00011	0.0020 ± 0.0007	3.300	0.0015 ± 0.0004	0.0009 ± 0.0007	1.667	0.0057 ± 0.0005	0.2375 ± 0.0387	0.024	0.0262 ± 0.0026	0.0125 ± 0.0003	2.096

N/D—not defined (have not been measured).

## Data Availability

The datasets used and analysed during the current study are available from the corresponding author upon reasonable request.

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
