# Peer review of "Multi-Omics Profiling of Human Endothelial Cells from the Coronary Artery and Internal Thoracic Artery Reveals Molecular but Not Functional Heterogeneity"

_ijms, 2023, doi:10.3390/ijms241915032_

Round 1

Reviewer 1 Report

This manuscript describes proteomic and transcriptomic profiling of endothelial cells derived from human coronary artery and human internal thoracic artery under static and flow conditions.

The manuscript has presented new data relating to different pathways and processes at the protein and mRNA level, however, no individual genes have been shown to be expressed. The manuscript needs a final proof of concept experiment comparing expression of a select number of genes for the two cell types at both static and flow culture. In addition, comparison of expression of individual key proteins at static vs flow culture is not presented

Other points to note

line 86 - give the full names for HCAEC and HITAEC the first time they are mentioned in the main body of the text

introduction appears to give a relatively detailed synopsis of the results, instead suggest that you give a clear hypothesis and then summarise the results at the start of the discussion section

Results Table 4 - the column headings that read "proteins" should be "genes", similarly for Table 5, Table 6, Table 7, Table 8

Author Response

We sincerely thank the reviewer for the constructive criticism and valuable notes, which collectively helped us to improve the paper. Please see the attachment.

Reviewer 2 Report

The authors state in the manuscript titled Multi-omics profiling of human endothelial cells from coronary artery and internal thoracic artery reveal molecular but not functional heterogeneity that suggest synergistic interactions between HCAEC and HITAEC in a bypass surgery scenario. However, there are some concerns that should be explained by the authors.

In the material and methods section, the authors should better explain the information. The detailed protocol could be considered as supplementary material.

According to the results the authors performed comparative studies between HCAEC and HITAEC, considering whole-transcriptome analysis and proteomic profiling. However, the study design shows some bias:

1.-Proteomic analysis considered two conditions (flow and static cell culture). The culture condition was not detailed.

2.- The authors performed studies based on gene ontology (GO) terms (BP, MF and CC). However, the results (Top 5 GO terms) show different features of functions for transcriptomic (Table 1-3) and proteomic (Flow condition: Table 4-6; Static condition: Table 7-9) analysis between HCAEC and HITAEC. The information in this section shows independent results, in which the specificity of the findings related to the scope shown by the authors is unclear. The authors stated in introduction section that HCAEC and HITAEC can be considered as a key factor to achieve a favorable outcome of CABG surgery in a long term. Therefore, the favorable outcome of CABG surgery goes beyond of the manuscript contribution. Furthermore, the authors did not perform studies of differential expression to RNA and protein level. There are not gene and protein annotations as well as absence of molecular interactions. The authors conclude that their results can be considered as a molecular basis of synergistic effect, but they only show studies based on  GO terms. The manuscript is largely lacks in data analysis and contribution.

Minor editing of English language required

Author Response

(The authors gave the same response as above.)

Reviewer 3 Report

The present article “Multi-omics profiling of human endothelial cells from coronary artery and internal thoracic artery reveals molecular but not functional heterogeneity” is interesting and novel.  I really appreciate the author’s thoughts and efforts.

Author Response

We sincerely thank the reviewer for the high evaluation of our study.

Round 2

Reviewer 1 Report

The manuscript is greatly improved by the addition of data (Supplementary table 25) and Figure 7.

I would consider moving supplementary table 25 to the main body of the manuscript as this sheds important light on the pathways analysis.

The legend for Figure 7 does not describe the figure clearly enough. I would suggest using A, B etc. for each pair of blots and the wording explaining the purpose of the lower blots is confusing.  

The data in Figure 8 does not seem to align to the rest of the manuscript, would consider removing this to the supplementary data or presenting some relevant molecular biomarkers in these patients

some minor editing needed but overall the quality of English is good

Author Response

(The authors gave the same response as above.)

Reviewer 2 Report

The authors have considered some concerns stated by the reviewer. However, there are some points did not board in the updated manuscript.

*According to the results, the authors performed comparative studies between HCAEC and HITAEC considering whole-transcriptome analysis (flow and static cell culture) and proteomic profiling (static cell culture). Why are there different conditions to Proteomic and transcriptome analysis?

*The Venn diagram could be detailed in text instead of large figures.

*According to the title, carried out experiments and approach shown by the authors. Features of functions detailed by GO terms are additional data, but not main information. All GO results should be in supplementary material. The authors should show as main results differential expression to RNA and protein level. There are not interaction networks for gene or protein profiling to understand better the relationship between HCAEC and HITAEC. The manuscript is largely lacks in data analysis and contribution.

Author Response

(The authors gave the same response as above.)

Round 3

Reviewer 2 Report

In this new version, the authors have addressed most of the concerns raised by the reviewer, expanded the results and modified the discussion according to the stated queries.